# Remote Assessment of Quality of Life and Functional Exercise Capacity in a Cohort of COVID-19 Patients One Year after Hospitalization (TELECOVID)

**DOI:** 10.3390/jcm11040905

**Published:** 2022-02-09

**Authors:** Yann Combret, Geoffrey Kerné, Flore Pholoppe, Benjamin Tonneville, Laure Plate, Marie-Hélène Marques, Helena Brunel, Guillaume Prieur, Clément Medrinal

**Affiliations:** 1Physiotherapy Department, Le Havre Hospital, F-76600 Le Havre, France; geoffrey.kerne@hotmail.fr (G.K.); flore.pholoppe37@gmail.com (F.P.); benjamin.tonneville@gmail.com (B.T.); laureisabelle.plate@ch-havre.fr (L.P.); gprieur.kine@gmail.com (G.P.); medrinal.clement.mk@gmail.com (C.M.); 2Pulmonology Department, Le Havre Hospital, F-76600 Le Havre, France; mariehelene.marques@ch-havre.fr; 3Saint Michel School of Physiotherapy, Paris-Saclay University, F-75015 Paris, France; helena.brunel@gmail.com; 4Research and Clinical Experimentation Institute (IREC), Pulmonology, ORL and Dermatology, Louvain Catholic University, 1200 Brussels, Belgium; 5Erphan, UVSQ, Paris-Saclay University, F-78000 Versailles, France

**Keywords:** COVID-19, functional exercise capacity, health-related quality of life, remote assessment

## Abstract

Studies have reported persistent symptoms in patients hospitalized for COVID-19 up to 6 months post-discharge; however, sequalae beyond 6 months are unknown. This study aimed to investigate the clinical status of COVID-19 patients one year after hospital discharge and describe the factors related to poor outcomes. We conducted a single-center, prospective, cohort study of patients in Le Havre hospital (France) between 1 March 2020 and 11 May 2020. Baseline characteristics were collected from medical charts (including KATZ index and Clinical Frailty scale (CFS)), and a remote assessment was conducted 12 months after discharge. The main outcomes were the scores of the physical and mental components (PCS and MCS) of the Short-Form 36 (SF-36) and performance on the one-minute sit-to-stand test (STST1′). Scores <50% of the predicted values were considered as poor, and univariate and multivariate analyses were undertaken to investigate factors related to poor outcomes. Remote assessment was performed for 128 of the 157 (82%) eligible patients. Twenty-two patients were admitted to the intensive care unit (ICU), 45 to the intermediate care unit (IU), and 61 to the general ward (GW). Patients who spent time in ICU were more independent and younger. A large proportion of the sample had poor physical (30%) and mental health (27%) and a poor functional exercise capacity (33%) at the remote assessment. Higher levels of frailty at admission and hospital discharge were, respectively, associated with a higher risk of poor functional exercise capacity (StdOR 3.64 (95%CI 1.39–10.72); *p* = 0.01) and a higher risk of poor mental health (StdOR 2.81 (95%CI 1.17–7.45); *p* = 0.03). Long-term outcomes following hospitalization for COVID-19 infection may be negative for at least one year after discharge. Remote follow-up assessment could be highly beneficial for COVID-19 patients.

## 1. Introduction

The COVID-19 outbreak has led to a very large number of hospitalizations worldwide since December 2019 [1,2]. A small proportion of patients (5–15%) require intensive care; however, a large proportion require prolonged hospitalization, which can lead to immobility and isolation [3,4,5]. These patients may return home with a wide spectrum of sequelae [6,7]. Follow-up studies have reported that most patients have post-discharge and/or post-COVID symptoms, including short- and medium-term respiratory complications, as well as chronic fatigue, muscle weakness, and anxiety or depression, all of which may reduce their quality of life [8,9,10,11].

Remote consultations have become increasingly frequent for both COVID-19 patients and patients with other chronic conditions, mainly because of the repeated lockdowns and the risk of in-hospital contaminations [12,13,14]. Video and telephone consultations have been found to be satisfactory means of monitoring COVID-19 patients discharged home early, conducting follow-up consultations, or even providing telerehabilitation [11,15,16].

Several cohort studies of patients hospitalized for COVID-19 reported respiratory and systemic consequences up to 6 months after discharge home; however, the consequences of COVID-19 beyond 6 months are still unclear, and a longer-term evaluation of these patients is warranted [8,9,10].

The primary objectives of this study were to (1) describe one-year quality of life and functional exercise capacity and (2) identify the factors associated with long-term impairment of quality of life (mental and physical) and functional exercise capacity in COVID-19 survivors hospitalized during the first wave of the pandemic. The secondary objective was to determine the feasibility of remote long-term assessment of COVID-19 patients.

## 2. Materials and Methods

### 2.1. Study Design and Setting

We conducted an observational prospective cohort study in Le Havre hospital (France). This study was approved by the Comité de Protection des Personnes Sud-Ouest et Outre-Mer III (N 20.11.17.61709) and registered on clinicaltrials.gov (NCT04714138). This study was reported in accordance with the Strengthening the Reporting of Observational Studies in Epidemiology (STROBE) reporting checklist [17].

### 2.2. Participants

All adult patients (>18 years) admitted for SARS-CoV-2 infection between 1 March and 11 May 2020 were eligible. COVID-19 diagnosis was made on the basis of a positive reverse-transcription polymerase chain reaction (RT-PCR) test or lung computed tomography (CT) and clinical symptoms. Non-inclusion criteria were: (1) patients who died during hospitalization or after hospital discharge; (2) length of stay (LOS) < 24 h; (3) incidental asymptomatic SARS-CoV-2 infection, and (4) patients under guardianship.

### 2.3. Data Extraction

Patient characteristics and comorbidities and the details of their hospitalization were retrospectively collected from our institution medical records. The highest-care ward they were admitted to during their stay (general ward (GW), intermediate care unit (IU), or intensive care unit (ICU)), treatments received (oxygen supplementation, non-invasive or invasive respiratory supports, and medications), discharge destination, and referral for rehabilitation were recorded. Independence was rated using the KATZ index of independence in activities of daily living (ADL) and the Clinical Frailty Scale (CFS) at admission and at hospital discharge. The KATZ score rates 6 common ADL: bathing, dressing, toileting, transferring, continence, and feeding, with a total score ranging from 0 (very dependent) to 6 (independent) [18]. The CFS evaluates the overall level of fitness or frailty, with a score ranging from 1 (very fit) to 9 (terminally ill) [19,20].

### 2.4. Remote Assessment

Between January and April 2021 (i.e., 10 to 12 months after hospital discharge), all eligible patients were contacted by telephone by one of the investigators (YC, GK, FP, BT, GP, or CM). In accordance with the French legislation, the consent process was conducted in two parts: (1) acceptance of participation and (2) declaration of non-opposition to participation. During the initial telephone consultation, the participants were asked to complete and confirm the data extracted from their medical charts and were asked questions relating to their current level of independence in order to rate the KATZ index and the CFS.

A remote video consultation was then scheduled immediately or within the following days, according to the patient’s availability. This second consultation was undertaken using a video consultation system (Therap-e software, GCS Normand’e-santé^®^, Louvigny, France), during which health-related quality of life (HRQOL) was assessed using the Short-Form 36 (SF-36) survey, and functional exercise capacity was assessed using the one-minute sit-to-stand test (STST1′). The use of video allowed the investigator to ensure that the STST1′ was performed correctly.

### 2.5. Measurements and Outcomes

HRQOL was assessed using the French version of the SF-36 survey, which is composed of one item related to changes in health during the previous year and 35 items within 8 categories: (1) physical functioning, (2) role limitations due to physical health, (3) pain, (4) general health, (5) emotional well-being, (6) role limitations due to emotional problems, (7) vitality, and (8) social functioning. The total score ranges from 0 to 100, with a higher score indicating a better health status [21,22]. Two summary measures were aggregated: the physical health component score (PCS) (items 1 + 2 + 3 + 4) and the mental component score (MCS) (items 5 + 6 + 7 + 8). Scores below 50% of the predicted values were considered to indicate poor physical or mental health [23].

Functional exercise capacity was measured with the STST1′, conducted according to the protocol described by Ozalevli et al. [24]. Patients were asked to sit on a regular chair (height 46–48 cm) with their arms crossed over their chest to avoid using them for leverage or support. They were encouraged to stand up a few times before the test to familiarize themselves with the task and to allow the investigators to verify their safety during the test. The investigator then gave standardized instructions to stand up and sit down completely as many times as possible within one minute. No verbal encouragement was provided. The patients were informed that they could rest during the test, but that the timer would not be stopped. Finally, the investigators notified the patients when 15 s were left. Patients were considered to have a poor functional exercise capacity if the number of STST1′ repetitions was <50% of the predicted values (PV) reported by Strassman et al. [25].

### 2.6. Statistical Analyses

Data are reported as numbers (and/or percentages), means (standard deviation (SD)), and medians (interquartile ranges (IQR)). Baseline characteristics and in-hospital treatment were compared according to the highest-care ward (i.e., GW, IU, or ICU) the patients had been admitted to during their stay. The baseline characteristics and discharge and follow-up outcomes of the 3 groups were compared using the ANOVA, Kruskal–Wallis, or chi-square tests, according to the type of variable. Univariate logistic regressions were then undertaken for the dependent variables poor physical or mental health (PCS or MCS < 50%, respectively) and poor functional exercise capacity (STST1′ < 50%PV) including the independent variables baseline characteristics (age, sex, comorbidities, and independence at admission), treatments received (respiratory support, LOS on highest-care ward, total LOS, and referral for rehabilitation (yes/no)), and discharge outcomes (independence at discharge and discharge destination). Odds Ratios with 95% confidence intervals and standardized odds ratios (StdOR) (calculated by dividing continuous independent variables by two standard deviations for standardization) are reported [26]. A multivariate adjusted logistic regression model was then computed for each dependent variable including the statistically significant independent variables from the primary univariate analyses. Statistical analyses were carried out with GraphPad Prism v9 and R (R Foundation for Statistical Computing, Vienna, Austria), and a *p*-value < 0.05 was considered significant.

## 3. Results

### 3.1. Study Population

Among the 264 patients hospitalized in Le Havre hospital between 1 March and 11 May 2020 for SARS-CoV-2 infection, 157 were eligible. A total of 128 (82% of the eligible patients) patients were reached and included 104 (81%) patients who completed the SF-36 survey and 94 (73%) patients who completed the STST1′ (Figure 1). Twenty-two of these patients had been admitted to ICU, 45 to IU, and 61 to GW, and all the patients included were diagnosed on the basis of a positive RT-PCR test. The participants included were not systematically different from those who were excluded regarding age, gender, working status, comorbidities, KATZ index and CFS at admission, highest-care ward they were admitted to, total hospital length of stay, and KATZ index and CFS at discharge; an exception was a higher BMI at admission (median BMI 27.6 vs. 24.9 kg.m^2^ in the included and excluded patients, respectively; *p* = 0.025).

### 3.2. Patients’ Characteristics

Patient characteristics are described in Table 1. Patients in the ICU group were younger (median age 64.5 (18) vs. 72 (18.5) years, *p* = 0.004) and had a higher baseline level of independence compared to those in the GW group. Patients in the ICU group had a longer LOS stay in the highest-care ward and a longer total LOS compared to the IU and GW groups (median duration: 11.5 (11) days vs. 6 (5.5) and 9 (6.5) days, and 17.5 (9.5) days vs. 7 (7) and 9 (6.5) days; *p* = 0.002 and *p* < 0.001, respectively). The level of independence was no longer different between groups at discharge. There was no difference in the proportion of patients referred for center-based rehabilitation between the three groups, but a larger proportion of the ICU group were prescribed home-based rehabilitation (41% vs. 4 and 7%, *p* < 0.001).

### 3.3. Remote Assessment

At the time of the remote consultation, the patients in the ICU group had a higher level of independence than those in the other two groups (higher scores for both the KATZ index and the CFS) (Table 2). PCS and MCS were 65% (38) and 77% (37), respectively, at the time of the consultation, 31 out of the 104 patients who completed the SF-36 survey had poor physical health (PCS < 50%), and 28 had poor mental health (MCS < 50%). The median number of STST1′ repetitions was 23 (IQR 22), corresponding to 67% PV (IQR 58), and almost one-third (31/94) of the patients had a poor functional exercise capacity (<50% PV). There was no significant difference in functional exercise capacity between the groups (Table 2).

### 3.4. Results of the Logistic Regressions

In the univariate logistic regression analyses, independent variables relating to independence at admission and discharge, LOS in the highest-care ward and total LOS, and referral for center-based rehabilitation were associated with poorer physical and mental health and poorer functional exercise capacity (Table 3). The highest-care ward admitted to (i.e., ICU, IU, or GW) was not associated with any of the dependent variables. In the multivariate logistic regression models, none of the independent variables were associated with the risk of poor physical health. However, higher ratings on the CFS (i.e., a higher level of frailty) at admission and discharge were, respectively, associated with a higher risk of poor functional exercise capacity (StdOR 3.64 (95%CI 1.39–10.72); *p* = 0.01) and a higher risk of poor mental health (StdOR 2.81 (95%CI 1.17–7.45); *p* = 0.03). In-hospital oxygen requirement was associated with a lower risk of poor functional exercise capacity (StdOR 0.20 (95%CI 0.05–0.68); *p* = 0.02) (Table 4).

## 4. Discussion

The results of this prospective cohort study of 128 patients evaluated 10 to 12 months after hospitalization for COVID-19 revealed that: (1) about one-third of the patients had poor long-term HRQOL and functional exercise capacity; (2) being admitted to a high-care ward during their stay (ICU or IU) did not appear to influence these outcomes; (3) a higher level of frailty at admission or at discharge was associated with a higher risk of long-term poor mental health and poor functional exercise capacity; (4) remote consultation was highly feasible for this population.

Remote consultation was achieved in a high proportion (82%) of the eligible patients. Furthermore, it was possible to measure important follow-up outcomes such as HRQOL and functional exercise capacity during a video consultation. Another study involving a telephone follow-up assessment four months after hospital discharge also reported a similar, small percentage of eligible patients who could not be reached (12% vs. 18% in the present study). This high proportion of patients followed up several months after hospital discharge is sufficient to ensure both a representative sample for research purposes and the clinical follow-up of a large number of patients [11]. Remote assessment is particularly relevant in the current context of repeated lockdowns and avoids the risk of in-hospital contaminations. Furthermore, remote assessment could provide a useful solution to the frequent perturbations of hospital organization and the scheduling of consultations due to the pandemic.

We found that the remote assessment of functional exercise capacity using the STST1′ was safe. The choice of the STST1′ was driven by the fact that this test is self-paced and thus has a low risk of adverse events and that it has been proposed as a first-line assessment for COVID-19 patients outside acute care services [27]. Furthermore, telehealth has been shown to be a feasible and efficient means to provide rehabilitation interventions, including home aerobic reconditioning and muscle strengthening exercises, for COVID-19 patients after discharge home [16,28]. In two studies, the intensity of the rehabilitation exercises was adapted according to self-reported symptoms or dyspnea, and no adverse effects were reported among the 38 patients who completed the programs [16,28]. The use of telehealth for both assessment and rehabilitation could therefore effectively compensate for the lack of organized care provision following hospital discharge and could be fully integrated into the care pathways of COVID-19 patients. In the present study, 28% of the patients included were referred to the rehabilitation unit within our hospital. This was the maximum capacity that this temporary unit could take during the pandemic. Other patients with significant impairment and low levels of independence were referred elsewhere for rehabilitation, but places were few due to the pandemic situation, and not all patients received the rehabilitation they required. Telerehabilitation after discharge was not undertaken in our center but could have been a useful alternative for patients who were well enough to be discharged home but who required further rehabilitation.

The importance of providing long-term rehabilitation for these patients was further demonstrated by the high prevalence of long-term poor physical and mental health and poor functional exercise capacity found in the present study sample (30%, 27%, and 33%, respectively). Although these outcomes were not evaluated at baseline, patients with COVID-19 who are at a high risk of hospitalization usually have significant comorbidities and thus likely already have a lower HRQOL and exercise capacity [4,8]. However, comparison of the CFS at admission and discharge showed that frailty was higher at discharge, although patients’ clinical status had improved. Thus, despite the fact all patients received rehabilitation in accordance with the current international guidelines, throughout their hospital stay, their frailty increased in hospital, regardless of the highest-care ward they were admitted to [29]. This is of particular importance, since our results revealed that frailty at admission and discharge were independently associated with long-term poor mental health and functional exercise capacity. Frailty was previously linked to adverse outcomes in COVID-19 patients and used for risk stratification, but this study is the first to link frailty to long-term outcomes [30,31]. Going further, the dynamics of frailty and independency were different among the three groups, and ICU patients recovered a larger level of fitness and daily life independency (KATZ score), despite the persistent limitations that were described in this study. These findings are consistent with those described for another cohort of COVID-19 ICU survivors that reported a median CFS score of 2, one year after discharge [32]. However, recovery in older patients seems to be much more challenging, since the KATZ score did not improve between discharge and follow-up in our GW group. Acute illness in older adults was previously shown as a high-risk factor for impairing ADL, including in the long term [33]. This was also recently described in COVID-19 older adults, with a significant proportion of patients (1 out of 3) reporting worsened ability to carry out activities of daily living 6 months after hospital discharge [34].

Patients hospitalized for COVID-19 have been shown to have subsequent moderate or severe muscle weakness, dyspnea, and fatigue, at least in the short term [16,35,36]. Moreover, a study by Martin et al. found a high prevalence of exercise limitation at hospital discharge: none of the 48 COVID-19 patients examined in that study reached the 50th percentile of the reference values for the STST1′ [16]. The results of the present study showed that a significant proportion of patients still presented exercise limitation one year after hospital discharge. Another point to note is that, despite the exercise limitation, the patients reported only moderate dyspnea (median 3 on the Borg scale), which suggests that their exercise capacity was not specifically limited by dyspnea. A similar level of dyspnea was found in a cross-sectional study by Paneroni et al. at the end of the STST1′ in COVID-19 patients evaluated at hospital discharge [35]. We propose three hypotheses to explain these results. Firstly, the STST1′ does not necessarily induce a high level of dyspnea. Secondly, the number of STST1′ repetitions may have been affected by muscle weakness (e.g., quadriceps weakness) in both the COVID-19 patients in the present study and the patients in the study by Paneroni et al. Thirdly, many COVID-19 patients report invalidating chronic fatigue that leads to early exercise cessation and significant discomfort during exercise. Unfortunately, this outcome was not formally measured in the present study, but fatigue was a major complaint of patients during the SF-36 survey and at the end of the STST1′.

Finally, the type of ward the patients were initially admitted to (i.e., ICU, IU, or GW) was not associated with poorer long-term outcomes. Other cohort studies reporting outcomes at 4 to 6 months found inconclusive results related to the impact of the type of ward patients were admitted to [8,11]. It is well known that a stay in ICU is related to a higher prevalence of muscle weakness and exercise limitation at hospital discharge [6,36,37]. However, the patients admitted to ICU in our sample were younger and had a lower level of frailty at baseline, suggesting that the poor outcomes observed at discharge were reversible, in contrast with what observed for patients with previous frailty. Furthermore, patients discharged from ICU were more frequently prescribed home rehabilitation, and closer monitoring was systematically organized after hospital discharge (as part of usual care). Both these factors may have contributed to the improved recovery of these patients.

The present study has several limitations. Firstly, it was uncontrolled, and the two main outcomes (HRQOL and functional exercise capacity) were not assessed at baseline. These two issues prevent firm conclusions being drawn regarding the impact of hospitalization for COVID-19 on these outcomes. Secondly, this study was single-center, thus the external validity of our findings may be moderate. Moreover, this study took place during the first wave of the pandemic, at a time when the management of COVID-19 patients was very heterogeneous. Thirdly, even though remote assessment was feasible for a large proportion of the study cohort, not all patients were able to complete both the SF-36 survey and the STST1′ (mostly due to the presence of cognitive disorders, difficulty understanding the instructions, or difficulty communicating), which could have impacted the results. Finally, one of the main limitations of the present study is the small study group, with an important imbalance between the study groups, which made a comparison between these groups difficult and impeded the drawing of unanimous conclusions.

## 5. Conclusions

Physical and mental health and functional exercise capacity remained poor in COVID-19 patients 10 to 12 months after hospital discharge. Admission to ICU rather than to another type of ward did not seem to influence these outcomes. However, frailty at admission and at discharge were independently associated with poor mental health and functional exercise capacity. Provision should be made for the long-term assessment and adequate referral of patients with persistent symptoms. Controlled studies are still needed to fully determine the impact of COVID-19 on these long-term outcomes.

## Figures and Tables

**Figure 1 jcm-11-00905-f001:**
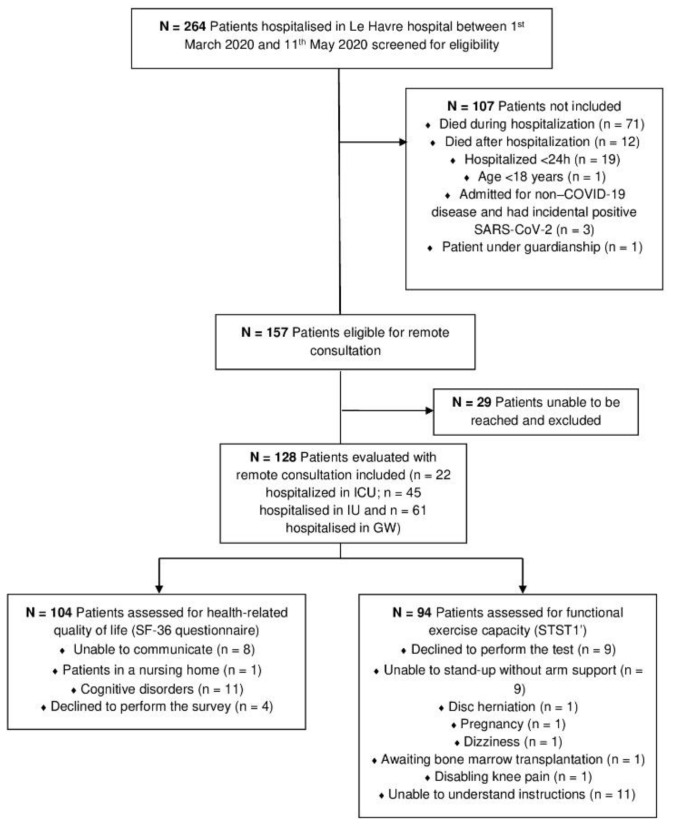
Flow diagram of patients screening and enrollment.

**Table 1 jcm-11-00905-t001:** Patient characteristics, details of hospital stay, and treatment on admission, during hospitalization, and at discharge.

	Complete Study Sample (N = 128)	Patients Discharged from ICU (N = 22)	Patients Discharged from IU (N = 45)	Patients Discharged from GW (N = 61)	*p*-Value ^‡^
Sex F/M, *n*	62/66	8/14	19/26	35/26	0.140
Age (years), median (IQR)	69 (20.8)	64.5 (18)	64 (23.5)	72 (18.5)	0.004
BMI (kg.m^2^), median (IQR)	27.6 (7.2)	29.8 (7.2)	27.7 (6.8)	27.3 (7.3)	0.057
Working status					0.046
Active, *n* (%)	44 (34)	8 (36)	21 (47)	15 (25)	
Retired, *n* (%)	80 (63)	14 (64)	21 (47)	45 (73)	
Unable to work, *n* (%)	4 (3)	0 (0)	3 (6)	1 (2)	
KATZ score at admission (0–6) *, median (IQR)	6 (0)	6 (0)	6 (0)	6 (2)	0.003
Clinical Frailty Scale at admission (1–9), median (IQR)	2.5 (2)	2 (1)	3 (2.5)	3 (4)	0.017
Comorbidities					
Chronic respiratory failure, *n* (%)	16 (13)	2 (9)	7 (16)	7 (11)	0.713
Chronic cardiac failure, *n* (%)	70 (55)	11 (50)	21 (47)	38 (62)	0.248
Diabetes, *n* (%)	29 (23)	9 (41)	5 (11)	15 (25)	0.021
Obesity, *n* (%)	41 (32)	11 (50)	14 (31)	16 (26)	0.121
Other, *n* (%) ****	42 (32)	7 (32)	14 (31)	21 (34)	0.932
Length of stay in highest-care ward admitted to (days), median (IQR) ***	8 (7)	11.5 (11)	6 (5.5)	9 (6.5)	0.002
Total hospital length of stay (days), median (IQR)	9 (10.5)	17.5 (9.5)	7 (7)	9 (6.5)	<0.001
Respiratory support					
Low-flow oxygen, *n* (%)	92 (72)	21 (95)	28 (62)	43 (70)	0.017
High-flow oxygen, *n* (%)	14 (11)	13 (59)	1 (2)	0 (0)	<0.001
CPAP/NIV, *n* (%)	4 (3)	4 (18)	0 (0)	0 (0)	<0.001
Invasive MV, *n* (%)	14 (11)	14 (64)	0 (0)	0 (0)	<0.001
Referral to the hospital rehabilitation unit, *n* (%)	34 (27)	5 (23)	9 (20)	20 (33)	0.306
Length of stay in rehabilitation unit (days), median (IQR)	12 (7.5)	10 (11.5)	14 (11)	12 (8)	0.677
KATZ score at discharge (0–6), median (IQR)	6 (2)	6 (2)	6 (1)	6 (4.5)	0.074
Clinical Frailty Scale at discharge (1–9), median (IQR)	4 (3)	4 (2)	4 (2)	5 (4)	0.246
Discharged home, *n* (%)	117 (91)	21 (95)	43 (96)	53 (87)	0.220
Home-based rehabilitation, *n* (%)	15 (12)	9 (41)	2 (4)	4 (7)	<0.001
Number of rehabilitation sessions, median (IQR)	20 (70)	10 (22)	70 (100)	57.5 (122)	0.140
Discharged to other rehabilitation centers, *n* (%)	11 (9)	2 (9)	2 (4)	7 (11)	0.441
Length of stay in rehabilitation center (days), median (IQR)	81 (110)	72 (95)	90 (260)	28 (110)	0.346

Abbreviations: BMI: body mass index; CPAP: continuous positive airway pressure; GW: general ward; ICU: intensive care unit; IQR: interquartile range; IU: intermediate care unit; MV: mechanical ventilation; NIV: non-invasive ventilation.* KATZ score ranges from 0 (very dependent) to 6 (independent) for six common ADL: bathing, dressing, toileting, transferring, continence, and feeding ** Other comorbidities include neurological diseases (stroke, Parkinson’s disease, poliomyelitis, multiple sclerosis), cancer, cognitive disorders, and amputations *** Length of stay in the initial ward indicates ICU length of stay for the ICU group (excluding the time spent in IU or GW after ICU discharge), IU length of stay for the IU group (excluding the time spent in GW after IU discharge), and the time spent in GW for the GW group; ^‡^
*p*-values are reported for the comparisons of the three groups of patients according to the highest-care ward they were admitted to, i.e., ICU vs. IU vs. GW.

**Table 2 jcm-11-00905-t002:** Results of the remote assessment.

	Complete Study Sample (N = 128)	Patients Discharged from ICU (N = 22)	Patients Discharged from IU (N = 45)	Patients Discharged from GW (N = 61)	*p*-Value ^‡^
KATZ score at one year (0–6) *, median (IQR)	6 (0)	6 (0)	6 (0)	6 (5)	<0.001
Clinical Frailty Scale at one year (0–6), median (IQR)	3 (3)	2 (1)	3 (2)	4 (5)	0.006
Number of STST1′ repetitions ^§^, median (IQR)	23 (22)	29 (20)	23 (18)	20 (29)	0.090
Number of STST1′ repetitions ^§^ (%predicted value), median (IQR)	67 (58)	84 (35)	61 (50)	61 (83)	0.089
End STST1′ dyspnea ^§^ (Borg 0–10), median (IQR)	3 (2)	3 (2)	4 (2)	3 (2)	0.211
SF-36 sub scores ^¶^					
Physical functioning (%), median (IQR)	65 (35)	68 (50)	70 (38)	65 (54)	0.903
Role limitations due to physical health (%), median (IQR)	75 (75)	62.5 (100)	75 (75)	63 (100)	0.177
Pain (%), median (IQR)	78 (55)	70 (53)	80 (65)	78 (55)	0.837
General Health (%), median (IQR)	60 (35)	65 (23)	55 (36)	50 (38)	0.268
Emotional well-being (%), median (IQR)	72 (28)	80 (17)	74 (25)	66 (32)	0.081
Role limitations due to emotional problems (%), median (IQR)	100 (77)	100 (100)	100 (77)	100 (77)	0.879
Vitality (%), median (IQR)	50 (29)	50 (23)	50 (33)	50 (24)	0.975
Social functioning (%), median (IQR)	100 (38)	100 (41)	88 (50)	100 (25)	0.378
Health change (%), median (IQR)	50 (25)	50 (31)	50 (25)	25 (25)	0.061
SF-36 Physical health component score(%), median (IQR)^¶^	65 (38)	68 (46)	68 (37)	63 (49)	0.789
SF-36 Mental health component score (%), median (IQR)^¶^	77 (37)	79 (41)	69 (33)	77 (39)	0.884
Returned to work at one year/active patients, *n* (%)	34/49 (69)	5/9 (56)	18/24 (75)	11/16 (69)	0.557

Abbreviations: GW: general ward; ICU: intensive care unit; IQR: interquartile range; IU: intermediate care unit; SF-36: short-form 36 questionnaire; STST1′: one-minute sit-to-stand test. * KATZ score ranges from 0 (very dependent) to 6 (independent) for six common ADL: bathing, dressing, toileting, transferring, continence, and feeding; ^‡^
*p*-values are reported for the comparisons of the three groups of patients according to the highest-care ward they were admitted to: ICU vs. IU vs. GW. ^§^ The results displayed for the STST1′ performance refer to the 94 patients that performed the test during the remote consultation. ^¶^ The results displayed for the SF-36 refer to the 104 patients that completed the survey during the remote consultation.

**Table 3 jcm-11-00905-t003:** Univariate logistic regression analyses for health-related quality of life and functional exercise capacity at the time of the remote assessment.

	Poor Physical HRQOL (PCS < 50% Predicted Value)			Poor Mental HRQOL (MCS < 50% Predicted Value)			Poor Functional Exercise Capacity (STST1′ < 50% Predicted Value)		
	OR (95%CI)	Std OR (95%CI)	*p*-Value	OR (95%CI)	Std OR (95%CI)	*p*-Value	OR (95%CI)	Std OR (95%CI)	*p*-Value
Age	1.04 (1–1.07)	1.74 (1.07–3)	0.03	1.02 (0.99–1.06)	1.40 (0.87–2.36)	0.18	1.02 (0.99–1.06)	1.42 (0.9–2.27)	0.14
Sex	1.87 (0.8–4.56)	1.87 (0.8–4.56)	0.16	1.47 (0.61–3.62)	1.47 (0.61–3.62)	0.39	1.44 (0.62–3.36)	1.44 (0.62–3.36)	0.40
BMI	0.96 (0.89–1.03)	0.77 (0.48–1.19)	0.26	0.96 (0.88–1.03)	0.77 (0.47–1.2)	0.27	0.97 (0.91–1.04)	0.85 (0.56–1.29)	0.45
KATZ at admission	0.33 (0.02–0.85)	0.14 (0–0.75)	0.14	0.81 (0.44–1.51)	0.70 (0.23–2.07)	0.46	-		
CFS at admission	1.95 (1.34–3.01)	3.42 (1.71–7.6)	0.001	1.32 (0.93–1.89)	1.66 (0.87–3.21)	0.12	1.93 (1.31–3.08)	3.36 (1.64–7.92)	0.002
Chronic Cardiac Insufficiency	2.69 (1.13–6.73)	2.69 (1.13–6.73)	0.03	1.41 (0.59–3.42)	1.41 (0.59–3.42)	0.44	2.04 (0.88–4.86)	2.04 (0.88–4.86)	0.10
Chronic Respiratory Insufficiency	1.21 (0.35–3.77)	1.21 (0.35–3.77)	0.75	2.03 (0.62–6.29)	2.03 (0.62–6.29)	0.22	2.18 (0.61–10.26)	2.18 (0.61–10.26)	0.26
Diabetes	2.20 (0.83–5.77)	2.20 (0.83–5.77)	0.11	2.10 (0.77–5.58)	2.10 (0.77–5.58)	0.14	1.65 (0.59–5.07)	1.65 (0.59–5.07)	0.35
Obesity	0.74 (0.29–1.8)	0.74 (0.29–1.8)	0.52	0.57 (0.2–1.46)	0.57 (0.2–1.46)	0.26	0.96 (0.39–2.41)	0.96 (0.39–2.41)	0.93
Service	0.97 (0.55–1.67)	0.97 (0.55–1.67)	0.90	0.91 (0.51–1.6)	0.91 (0.51–1.6)	0.76	0.71 (0.41–1.24)	0.71 (0.41–1.24)	0.23
LOS in initial ward	1.08 (1.01–1.16)	2.37 (1.17–5.18)	0.02	1.11 (1.04–1.2)	3.21 (1.53–7.58)	0.004	1.03 (0.96–1.11)	1.36 (0.66–3.14)	0.43
Total hospital LOS	1.04 (0.99–1.1)	1.59 (0.86–2.99)	0.14	1.06 (1.01–1.13)	2.10 (1.12–4.15)	0.03	0.99 (0.94–1.05)	0.91 (0.49–1.75)	0.78
Referral to the hospital rehabilitation unit	7.76 (2.8–23.45)	7.76 (2.8–23.45)	<0.001	3.28 (1.2–9.05)	3.28 (1.2–9.05)	0.02	3.14 (1.04–11.73)	3.14 (1.04–11.73)	0.06
Rehabilitation unit LOS	1.16 (1.07–1.28)	3.66 (1.81–8.62)	0.001	1.11 (1.04–1.21)	2.52 (1.35–5.16)	0.01	1.15 (1.03–1.34)	3.30 (1.34–12.83)	0.03
Oxygen requirement	1.01 (0.4–2.75)	1.01 (0.4–2.75)	0.98	2.60 (0.88–9.6)	2.60 (0.88–9.6)	0.11	0.25 (0.07–0.75)	0.25 (0.07–0.75)	0.02
High-flow oxygen requirement	1.37 (0.39–4.36)	1.37 (0.39–4.36)	0.60	1.62 (0.46–5.2)	1.62 (0.46–5.2)	0.43	0.37 (0.1–1.26)	0.37 (0.1–1.26)	0.11
Invasive MV	0.93 (0.24–3.07)	0.93 (0.24–3.07)	0.91	1.62 (0.46–5.2)	1.62 (0.46–5.2)	0.43	0.45 (0.12–1.61)	0.45 (0.12–1.61)	0.21
KATZ at discharge	0.62 (0.43–0.85)	0.38 (0.18–0.73)	0.01	0.60 (0.41–0.83)	0,35 (0.17–0.68)	0.003	0.56 (0.29–0.84)	0.30 (0.08–0.71)	0.02
CFS at discharge	1.81 (1.29–2.64)	2.76 (1.55–5.25)	0.001	2.20 (1.51–3.36)	3.84 (2.03–7.97)	<0.001	1.41 (1.03–1.97)	1.79 (1.06–3.19)	0.04
Discharged Home	0.19 (0.03–1.03)	0.19 (0.03–1.03)	0.06	0.34 (0.06–1.95)	0.34 (0.06–1.95)	0.21	0.83 (0.11–4.51)	0.83 (0.11–4.51)	0.84
Home-based rehabilitation	2.68 (0.77–9.34)	2.68 (0.77–9.34)	0.11	3.18 (0.91–11.17)	3.18 (0.91–11.17)	0.06	0.99 (0.23–5.07)	0.99 (0.23–5.07)	0.99
Number of rehabilitation sessions	1.01 (1–1.04)	1,39 (0.91–2.57)	0.17	1.02 (1–1.04)	1.44 (0.94–2.71)	0.14	1.05 (1–1.36)	3,08 (0.92–1114)	0.48
Discharged to other Rehabilitation Center	8.52 (1.83–60.8)	8.52 (1.83–60.8)	0.01	3.00 (0.66–13.59)	3.00 (0.66–13.59)	0.14	0.99 (0.23–5.07)	0.99 (0.23–5.07)	0.99
Rehabilitation Center stay length	1.02 (1–1.05)	1.97 (1.04–5.33)	0.07	1.02 (1–1.05)	1.99 (1.06–5.09)	0.05	1.00 (0.98–1.03)	1.14 (0.57–3.05)	0.73

Abbreviations: BMI: body mass index; CFS: clinical frailty scale; HRQOL: health-related quality of life; LOS: length of stay; MCS: mental component score; MV: mechanical ventilation; OR: odds ratio, PCS: physical component score; StdOR: standardized odds ratio; STST1′: one-minute sit-to-stand test.

**Table 4 jcm-11-00905-t004:** Multiple regression models for health-related quality of life and functional exercise capacity at the time of the remote assessment.

	Poor Physical HRQOL (PCS < 50% Predicted Value)			Poor Mental HRQOL (MCS < 50% Predicted Value)			Poor Functional Exercise Capacity (STST1′ < 50% Predicted Value)		
	Std OR	(95% CI)	*p*-Value	Std OR	(95% CI)	*p*-Value	Std OR	(95% CI)	*p*-Value
Age	0.87	(0.45–1.69)	0.68	-			-		
CFS at admission	2.10	(0.86–5.65)	0.12	-			3.64	(1.39–10.72)	0.01
Chronic Cardiac Insufficiency	1.85	(0.55–6.48)	0.32	-			-		
Service LOS	0.98	(0.36–2.58)	0.96	1.66	(0.23–1.7)	0.26	-		
Referral to the hospital rehabilitation unit	2.09	(0.24–16.59)	0.48	0.35	(0.7–4.27)	0.40	-		
Rehabilitation unit LOS	1.48	(0.45–6.15)	0.53	2.50	(0.68–15.65)	0.24	2.33	(0.9–10.95)	0.17
Oxygen Requirement	-			-			0.20	(0.05–0.68)	0.02
KATZ at discharge	0.75	(0.25–2.11)	0.60	0.92	(0.36–2.32)	0.86	0.36	(0.07–1.21)	0.14
CFS at discharge	1.34	(0.49–3.82)	0.58	2.81	(1.17–7.45)	0.03	0.67	(0.26–1.66)	0.39
Discharged to other rehabilitation Center	4.07	(0.56–38.37)	0.18	-			-		

Abbreviations: CFS: clinical frailty scale; CI: confidence interval; HRQOL: health-related quality of life; LOS: length of stay; MCS: mental component score; OR: odds ratio; PCS: physical component score; StdOR: standardized odds ratio; STST1′: one-minute sit-to-stand test.

## Data Availability

Yann Combret had full access to all the data in the study and takes responsibility for the integrity of the data and the accuracy of the data analysis. Data will be made available from the corresponding author, upon reasonable request.

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
