# Peer review of "Remote Assessment of Quality of Life and Functional Exercise Capacity in a Cohort of COVID-19 Patients One Year after Hospitalization (TELECOVID)"

_jcm, 2022, doi:10.3390/jcm11040905_

Round 1
Reviewer 1 Report
Relevant study of one-year follow-up of COVID19 patients that were admitted to the Le Havre regional hospital, using teleconsultation. The study is well organized and planned and is methodologically sound. It gives information about the clinical status of COVID 19 patients at one year, and what factors contribute to the clinical status. It demonstrates the usefulness and feasibility of teleconsultation for follow-up and the possibility of making video-assisted tests.
Although it is written that diagnoses have been made using PCR or Lung CT aspects and clinical history, how many patients were diagnosed without PCR confirmation?
What happened to the patients that didn't participate in the follow-up teleconsultation (18%) and to those that rejected or couldn't be tested using the one-minute sit-to-stand test
Author Response
Relevant study of one-year follow-up of COVID19 patients that were admitted to the Le Havre regional hospital, using teleconsultation. The study is well organized and planned and is methodologically sound. It gives information about the clinical status of COVID 19 patients at one year, and what factors contribute to the clinical status. It demonstrates the usefulness and feasibility of teleconsultation for follow-up and the possibility of making video-assisted tests.
We thank the Reviewer for these positive comments and for acknowledging the clinical relevance of our work.
Although it is written that diagnoses have been made using PCR or Lung CT aspects and clinical history, how many patients were diagnosed without PCR confirmation?
We thank the Reviewer for this question. The concerned sentence in the Methods section was probably misleading. Actually, the criteria for COVID-19 diagnosis were set a priori in the protocol study and we chose to include patients with a COVID-19 diagnosis based on reverse transcription–polymerase chain reaction (RT-PCR) or lung computed tomography (CT) and clinical symptoms, according to previously published studies on the same topic.
However, none of the study sample was diagnosed on the basis of CT and clinical symptoms, and all the patients included had a positive RT-PCR test. We precised these two points in the Methods and Results section.
P2, L71: “COVID-19 diagnosis could be made on the basis of a positive reverse transcription–polymerase chain reaction (RT-PCR) test or lung computed tomography (CT) and clinical symptoms.”
P4, L152: “Twenty-two of these patients had been admitted to ICU, 45 to IU and 61 to GW, and all the patients included were diagnosed on the basis of a positive RT-PCR test.”
What happened to the patients that didn't participate in the follow-up teleconsultation (18%) and to those that rejected or couldn't be tested using the one-minute sit-to-stand test
We thank the Reviewer for this question.
The 29 patients that were not able to be reached for the purpose of this study were contacted according to the contact procedure that was protocolized a priori:
- A maximum of 5 calls per patient were attempted using the telephone number registered were tried
- If the phone number was not attributed or the patient did not answer any of the calls, the trusted person of each patient registered in our medical charts was contacted
- If none of these calls were successful, the patient was considered “unable to be reached”
For the description of our study sample, the following changes were made according to the comments of the Reviewers and the Academic Editor :
(1) Those 29 patients are finally not included in the study sample in this new version according to the comments of the Reviewer #2, and the final study sample encompasses 128 patients (vs. 157 patients previously)
(2) They are described for the purpose of selection bias description in the Results according to the Academic Editors comments
P4, L154: “The participants included were not systematically different from those who were excluded regarding age, gender, working status, comorbidities, KATZ index and CFS at admission, highest-care ward they were admitted to, total hospital length of stay and KATZ index and CFS at discharge; except for a higher BMI at admission (median BMI 27.6 vs 24.9 kg.m2 in the included and excluded patients respectively; p=0.025).”
Reviewer 2 Report
This is an interesting article about remote video consultation of COVID-19 patients one year after severe COVID-19 infection.
How did the authors retrospectively evaluate the Katz index and CSF at admission and discharge? Are all the elements of these questionnaires clearly identified and described in the source data?
The flow diagram is incorrect. 29 patients who could not be contacted should be excluded from the study and further analysis.
Table 1 is unclear. The research sample should be 128 points, not 157. It is not known what p means and which columns were compared. The KATZ result should be explained in the legend
Table 2 is unclear. Only 104 patients completed SF-36 and 94 patients completed STST and this number should be summarized in the table. It is not known what p stands for which columns were compared. The KATZ result should be explained in the legend.
Can the authors answer the question of whether 157 or 128 patients were included in the univariate and multivariate analysis?
There is a lack of KATZ scores in the discussion, only a few on CSF, and no comparison with other studies.
Author Response
This is an interesting article about remote video consultation of COVID-19 patients one year after severe COVID-19 infection.
We thank the Reviewer for this positive comment.
How did the authors retrospectively evaluate the Katz index and CSF at admission and discharge? Are all the elements of these questionnaires clearly identified and described in the source data?
We thank the reviewer for this interesting question.
The KATZ index was directly evaluated by the healthcare providers (mainly the nurses or the nurse assistants) at admission and at discharge. Each of the 6 items were evaluated, and the complete index was rated for all the patients. The scores were simply collected.
The Clinical Frailty Scale was rated for the purpose of the study by the investigators, on the basis of the medical charts. The complete medical observations at admission and at discharge were reviewed to rate the scale. Each information regarding the clinical status of the patient, his functional abilities and the level of potential assistance needed were collected.
Moreover, the two items were verified with the patient and/or his caregivers at the time of the first telephone contact or during the remote consultation, and any discrepancy was investigated for correction.
The flow diagram is incorrect. 29 patients who could not be contacted should be excluded from the study and further analysis.
We thank the Reviewer for helping us correcting this ambiguity. We aimed to describe the characteristics of all the patients hospitalized in Le Havre hospital during the first wave of the pandemic, but it was indeed senseless to consider the patients that were not reached out as “included” in the present study. The flow diagram of patient selection has been modified accordingly, excluding these 29 patients.
The characteristics of the patients that were not able to be reached and excluded are now precised for describing the selection bias in our study, according to the comment formulated by the Editor.
Table 1 is unclear. The research sample should be 128 points, not 157. It is not known what p means and which columns were compared. The KATZ result should be explained in the legend
We thank the reviewer for this precision. The results displayed in the Table 1 were corrected according to the aforementioned exclusion of the 29 patients that were not able to be reached. Precisions have been added to the legend of the table to explain the comparisons that were undertaken and the results of the KATZ score.
P6, L185: “* KATZ score ranges from 0 (very dependent) to 6 (independent) for 6 common ADL: bathing, dressing, toileting, transferring, continence and feeding”
P6, L190: “‡ P-values are reported for the comparisons of the 3 groups of patients according to the highest-care ward they were admitted to: ICU vs. IU vs. GW”
The following changes were also computed for harmonization:
P1, L26: “Remote assessment was performed for 128 of the 157 (82%) eligible patients. Twenty-two patients were admitted to the intensive care unit (ICU), 45 to the intermediate care unit (IU) and 61 to the general ward (GW).”
P4, L150: “A total of 128 (82% of the eligible patients) patients were reached and included: 104 (81%) completed the SF-36 survey and 94 (73%) completed the STST1’ (Figure 1). Twenty-two of these patients had been admitted to ICU, 45 to IU and 61 to GW, and all the patients included were diagnosed on the basis of a positive RT-PCR test. The participants included were not systematically different from those who were excluded regarding age, gender, working status, comorbidities, KATZ index and CFS at admission, highest-care ward they were admitted to, total hospital length of stay and KATZ index and CFS at discharge; except for a higher BMI at admission (median BMI 27.6 vs 24.9 kg.m2 in the included and excluded patients respectively; p=0.025).”
P4, L173: “Patient characteristics are described in Table 1. Patients in the ICU group were younger (median age 64.5 (18) vs 72 (18.5) years, p=0.004) and had a higher baseline level of independence compared to those in the GW group. Patients in the ICU group had a longer LOS stay in the highest-care ward and a longer total LOS compared to the IU and GW groups (median duration: 11.5 (11) days vs. 6 (5.5) and 9 (6.5) days; and 17.5 (9.5) days vs. 7 (7) and 9 (6.5) days; p=0.002 and p<0.001 respectively). The level of independence was no longer different between groups at discharge. There was no difference in the proportion of patients referred for center-based rehabilitation between the three groups, but a larger proportion of the ICU group were prescribed home-based rehabilitation (41% vs. 4 and 7%, p<0.001).”
P9, L254: “The results of this prospective cohort study of 128 patients evaluated 10 to 12 months after hospitalization for COVID-19 revealed that:”
Table 2 is unclear. Only 104 patients completed SF-36 and 94 patients completed STST and this number should be summarized in the table. It is not known what p stands for which columns were compared. The KATZ result should be explained in the legend.
We thank the reviewer for this thoughtful comment. The number of patients evaluated using the STST1’ and the SF-36 survey are now precised in the legend of the Table 2. According to the previous comment, the comparisons undertaken and the results of the KATZ score are now explained in the legend of this table.
P7, L208: “* KATZ score ranges from 0 (very dependent) to 6 (independent) for 6 common ADL: bathing, dressing, toileting, transferring, continence and feeding ‡ P-values are reported for the comparisons of the 3 groups of patients according to the highest-care ward they were admitted to: ICU vs. IU vs. GW § The results displayed for the STST1’ performance refer to the 94 patients that performed the test during the remote consultation ¶ The results displayed for the SF-36 refer to the 104 patients that completed the survey during the remote consultation”
Can the authors answer the question of whether 157 or 128 patients were included in the univariate and multivariate analysis?
We thank the reviewer for this question. Actually, the 128 patients that were able to be reached were included in the univariate and multivariate analysis. This seems to reinforce the logic for excluding the other 29 patients, for whom remote assessment was not feasible. However, we remain fully available to discuss this particular point if the reviewer has other questions.
There is a lack of KATZ scores in the discussion, only a few on CSF, and no comparison with other studies.
We thank the reviewer for this comment. A literature review was conducted to integrate discussion elements concerning the measurements of overall fitness and independency investigated in our study. Numerous studies indeed used the Clinical Frailty Scale in the context of the COVID-19 pandemic and especially for the risk stratification of those patients. However, we have found recent evidence describing the level of fitness using the CFS in COVID-19 survivors, and the level of independency using the KATZ score items in older adults following hospitalization for COVID-19. These results are now integrated and compared to ours in the Discussion section of our manuscript.
P10, L305: “Frailty was previously linked with adverse outcomes in COVID-19 patients, and used for risk stratification, but this study is the first to link frailty and long-term outcomes [30, 31]. Going further, the dynamics of frailty and independency were different among the three groups, and ICU patients recovered a larger level of fitness and daily-life independency (KATZ score), despite the persistent limitations that were described in this study. These findings are consistent with another cohort of COVID-19 ICU survivors that reported a median CFS score of 2, one-year after discharge [32]. However, the recovery in older patients seem to be much more challenging, since KATZ score did not improve between discharge and follow-up in our GW group. Acute illness in older adults was previously showed as a high risk for impairing ADL, including in the long-term [33]. This was also recently described in COVID-19 older adults, with a significant proportion of patients (1 out of 3) reporting worsened ability to carry out activities of daily living, 6 months after hospital discharge [34].”
The following references have been integrated:
[30] Labenz C, Kremer WM, Schattenberg JM, Wörns M-A, Toenges G, Weinmann A, et al. Clinical Frailty Scale for risk stratification in patients with SARS-CoV-2 infection. J Investig Med. 2020 Aug;68(6):1199–202.
[31] Andrés-Esteban EM, Quintana-Diaz M, Ramírez-Cervantes KL, Benayas-Peña I, Silva-Obregón A, Magallón-Botaya R, et al. Outcomes of hospitalized patients with COVID-19 according to level of frailty. PeerJ. 2021 Apr 13;9:e11260.
[32] Heesakkers H, van der Hoeven JG, Corsten S, Janssen I, Ewalds E, Simons KS, et al. Clinical Outcomes Among Patients With 1-Year Survival Following Intensive Care Unit Treatment for COVID-19. JAMA [Internet]. 2022 Jan 24 [cited 2022 Jan 28]; Available from: https://jamanetwork.com/journals/jama/fullarticle/2788504
[33] Covinsky KE, Palmer RM, Fortinsky RH, Counsell SR, Stewart AL, Kresevic D, et al. Loss of independence in activities of daily living in older adults hospitalized with medical illnesses: increased vulnerability with age. J Am Geriatr Soc. 2003 Apr;51(4):451–8.
[34] Walle-Hansen MM, Ranhoff AH, Mellingsæter M, Wang-Hansen MS, Myrstad M. Health-related quality of life, functional decline, and long-term mortality in older patients following hospitalisation due to COVID-19. BMC Geriatr. 2021 Dec;21(1):199.
Round 2
Reviewer 2 Report
The authors took my comments into account. One of the main limitations of this study is the small study group. It is worth adding this in the limitations When the group include in ICU had (N = 22), in IU had (N = 45), and in GW had (N = 61) it is difficult to compare and draw unanimous conclusions.Author Response
We thank the Reviewer for this positive feedback, and for this relevant new comment. Indeed, this should be cited as a limit for the present work and is now mentioned in the concerned section.
The following sentence has been added to the Discussion section:
P11, L358: “Finally, one of the main limitations of the present study is the small study group, with an important imbalance between the study groups making these groups difficult to compare and impeding the drawing of unanimous conclusions.”
